# Surgery and Perioperative Management in Small Intestinal Neuroendocrine Tumors

**DOI:** 10.3390/jcm9072319

**Published:** 2020-07-21

**Authors:** Sophie Deguelte, Marine Perrier, Cheryne Hammoutene, Guillaume Cadiot, Reza Kianmanesh

**Affiliations:** 1Digestive Surgery Department, Reims University Hospital, Robert Debré Hospital, F-51092 Reims, France; chammoutene@chu-reims.fr (C.H.); rkianmanesh@chu-reims.fr (R.K.); 2Clinical Research Department, Reims University Hospital, Robert Debré Hospital, F-51092 Reims, France; 3Department of Hepato-Gastroenterology and Digestive Oncology, Reims University Hospital, Robert Debré Hospital, F-51092 Reims, France; mperrier@chu-reims.fr (M.P.); gcadiot@chu-reims.fr (G.C.)

**Keywords:** small bowel neuroendocrine tumor, surgery, carcinoid syndrome, carcinoid crisis

## Abstract

Small-intestinal neuroendocrine tumors (SI-NETs) are the most prevalent small bowel neoplasms with an increasing frequency. In the multimodal management of SI-NETs, surgery plays a key role, either in curative intent, even if R0 resection is feasible in only 20% of patients due to advanced stage at diagnosis, or palliative intent. Surgeons must be informed about the specific surgical management of SI-NETs according to their hormonal secretion, their usual dissemination at the time of diagnosis and the need for bowel-preserving surgery to avoid short bowel syndrome. The aim of this paper is to review the surgical indications and techniques, and perioperative and postoperative management of SI-NETs.

## 1. Introduction

Small-intestinal neuroendocrine tumors (SI-NETs) are the most prevalent small bowel neoplasms [1]. SI-NETs demonstrate a strong predilection for nodal and distant metastases; despite their relative slow progression, the five-year overall survival rates in patients with localized and metastatic disease are 70% to 100% and 35% to 60%, respectively [2,3,4]. Figure 1 summarizes SI-NET classical presentation. SI-NET primary tumors are usually small (<20 mm), distal in the ileum, and multiple in 30% to 50% of cases [4,5,6]. Mesenteric lymph node metastases (MLNM) are present in more than 80% of patients at diagnosis, regardless of the size of the primary tumor [7,8,9,10,11]. MLNM are typically larger than primary tumors and associated with dense desmoplastic fibrosis leading to retractile mesenteritis. When extending to the small bowel and the mesenteric vessels, MLNM can cause acute symptoms of abdominal pain, bowel obstruction, mesenteric ischemia, and digestive hemorrhage which may require emergency surgery in up to 25% of cases [5,12]. The liver is the most common metastatic site. Liver metastases are present in approximately 50% of patients followed by the peritoneal cavity; peritoneal carcinomatosis is present in approximately 20% of patients [10,11]. SI-NETs are also characterized by their ability to secrete vasoactive substances (such as serotonin, tachykinins, and prostaglandins) responsible for mesenteric fibrosis, carcinoid syndrome, carcinoid crisis, and carcinoid heart disease. These paraneoplastic manifestations usually affect patients in the context of extensive liver metastases which allow for bypassing of hormone release from portal circulation [13].

In the multimodal management of SI-NETs, surgery plays a key role, either in curative intent, even if R0 resection is feasible in only 20% of patients due to advanced stage at diagnosis, or palliative intent [14,15]. The aim of this paper is to review the surgical indications and techniques, and perioperative and postoperative management of SI-NETs.

## 2. Preoperative Imaging

Preoperative imaging in SI-NETs is time- and cost-consuming, requiring both morphological and metabolic imaging. Imaging work-up aims to map and stage the disease with a view to scheduling treatment even if tumor burden is often underestimated in all imaging modalities [16,17].

Morphological imaging provides anatomical information, such as the exact size and location of lesions. Morphological imaging techniques like contrast-enhanced computed tomography (CT) scans and magnetic resonance imaging (MRI) are routinely used in first line for either diagnosis, staging or follow-up [14].

Functional imaging provides whole-body imaging which facilitates the diagnosis of abdominal and extra-abdominal lesions. Functional imaging may also evaluate tumor aggressiveness and related therapeutic possibilities (for peptide-radionuclide receptor therapy (PRRT)). Three functional imaging techniques can be used in SI-NETs. The first one is SomatoSTatin Receptor Imaging (SSTRI) which has diagnostic value, a prognostic value high uptake is correlated with better survival and a theranostic value high uptake is correlated with PRRT efficacy [18,19]. SSTRI includes somatostatin receptor scintigraphy and CT (positron emission tomography (PET)/CT) with ^68^Gallium (^68^Ga-PET)-labeled somatostatin analogues (^68^Ga-DOTATOC, ^68^Ga-DOTATATE, ^68^Ga-DOTA-NOC). ^68^Ga-PET is now preferred, because of its considerably better yield for well-differentiated neuroendocrine tumor (NET) imaging [20]. The second one is 18F-fluorodihydroxyphenylalanine-positron emission tomography (FDOPA-PET) which has an excellent diagnostic accuracy compared with cross-sectional morphological imaging and somatostatin receptor scintigraphy, but it has been poorly compared with ^68^Ga-PET [21,22,23]. FDOPA-PET presents no theranostic role and its prognostic value should be explored. FDOPA-PET may be an alternative to ^68^Ga-PET for the preoperative evaluation in SI-NETs when ^68^Ga-PET is not available. The third one is 18F-2-fluorodeoxyglucose (FDG-PET) which mainly has a prognostic value high FDG uptake which is correlated with worse survival even in well-differentiated NETs such as SI-NETs. The results of FDG-PET and SSTRI can be combined to perform metabolic grading [24,25,26]. Metabolic grading has been reported to be better than the Ki67 proliferation index to predict tumor aggressiveness in SI-NET of any grade, by providing a whole-body evaluation of tumor aggressiveness and reducing the sampling bias.

### 2.1. Primary Tumor(s) Imaging

Primary SI-NETs may be non-visualized on imaging (morphological and/or functional) and only suspected in cases of typical MLNM with retractile mesenteritis or in cases of neuroendocrine liver metastases with high levels of urinary 5-hydroxyindoleacetic acid (5-HIAA) and in the absence of pulmonary or pancreatic tumors on a CT scan. Additional non-invasive techniques may improve the rate of primary tumor detection, such as CT or magnetic resonance (MR) enteroclysis, and capsule endoscopy. However, these additional imaging procedures seem useless for the preoperative work-up, as none of them can replace manual bowel palpation at the time of the surgery [16,27,28].

### 2.2. MLNM Imaging

For MLNM, the challenge is not the identification of the lesion but the assessment of its resectability. Surgeons should not consider the MLNM complex to be inextricable and, conversely, should not indicate a laparotomy for an inextricable lesion [29]. For a precise assessment, attention must be focused on the first jejunal arteries located along the right side of the mesenteric artery on both axial and frontal slices, on an early arterial phase contrast-enhanced abdominal-pelvic CT and/or MRI [29]. At least three free jejunal arteries have been arbitrarily deemed necessary to preserve a sufficient length of vascularized residual small bowel [29]. A classification based on the relationship between MLNM and the superior mesenteric vessels with a view to predicting the difficulties and the possibilities of this resection was previously proposed by our team (Figure 2) [29].

### 2.3. Imaging of Liver Metastases

Detection of liver metastases on metabolic imaging is not sufficient before liver surgery. Liver MRI, which is more effective than a CT scan for the detection of liver metastases [30], is usually required to characterize their resectability [30,31]. Liver MRI should include T2-weighted sequences and diffusion-weighted sequences which are more sensitive. Whatever the preoperative work-up, more metastases are detected on surgical exploration and even more on pathological examination. Since fewer than 50% of liver metastases are detected preoperatively, when micro-metastases (<1 mm) are discovered, they are nearly always present outside the macro-metastases in SI-NETs [32,33,34].

As defined by the preoperative assessment, liver metastases can be classified into three groups: one metastasis of any size (type I); one metastasis accompanied by smaller lesions in both lobes (type II); and disseminated metastatic spread with both liver lobes always involved (type III) [35].

### 2.4. Peritoneal Carcinomatosis Imaging

In SI-NETs, carcinomatosis and ovarian metastases are respectively found in 20% and 4% of patients [3]. As in other diseases, the sensitivity of CT and MRI is poor for the detection of peritoneal carcinomatosis [36]. Regarding the sensitivity of metabolic imaging, data are scarce but neither FDOPA-PET nor ^68^Ga-PET seem able to detect them with an accuracy of 100% as cited in small published case series [16,36]. To date, all existing imaging tools are unable to perform the preoperative assessment of peritoneal disease with precision.

## 3. Pre and Perioperative Management of the Hormonal Syndrome

The perioperative period must be exploited to prepare the patient for surgery, either by specific preparation for SI-NET surgery or by general preparation for oncological abdominal surgery. Specific preparation includes the diagnosis and management of carcinoid syndrome, and of carcinoid heart disease, and the prevention and treatment of carcinoid crisis. The general preparation includes immunonutritional support, optimization of the patient’s comorbidities, and correction of bad behavior (smoking cessation). The general preparation is not detailed in this review.

### 3.1. Diagnosis and Management of Carcinoid Syndrome

Carcinoid syndrome is characterized by episodic facial flushing (60–85%), diarrhea (60–80%), abdominal cramps, hypotension, and intermittent bronchial wheezing [13]. The preoperative prevalence of carcinoid syndrome is low at 32.4% in patients with SI-NETs [37], but acute and chronic complications from carcinoid syndrome may impact on overall survival and quality of life [38].

The diagnosis of carcinoid syndrome is confirmed by the dosage of 5-HIAA, a metabolite of serotonin, the main mediator of carcinoid syndrome with a sensitivity of up to 100% and specificity of 85% to 90% [39]. The urinary 5-HIAA level also assesses the severity of carcinoid syndrome. Thus, a 24-h urinary 5-HIAA measurement should be systematically performed preoperatively for both screening for carcinoid syndrome and predicting the risk of carcinoid heart disease. Studies aiming to validate the use of plasma 5-HIAA level, an easier alternative, as well as confirm and assess carcinoid syndrome, are in progress [40].

Chromogranin A is not recommended for the preoperative work-up, but most research teams measure it because it is often used for the follow-up.

Upon evidence or suspicion of carcinoid syndrome, antisecretory therapy with somatostatin analogues must be initiated before any invasive procedure in SI-NET patients [41]. There are currently no recommendations concerning doses of somatostatin analogues. Long-acting octreotide [42] could be administered intramuscularly at a dose of 20 to 30 mg every four weeks, and lanreotide [43] could be administered subcutaneously at a dose of 120 mg every four weeks.

### 3.2. Cardiac Evaluation

Carcinoid heart disease is the most feared complication of carcinoid syndrome. It is characterized by a fibrotic degeneration of the valves, particularly affecting the right heart chambers—isolated tricuspid valve regurgitation in up to 90% of cases—leading to a deterioration of the right ventricular function [38,44,45]. Carcinoid heart disease is detected in approximately 40% of patients with a carcinoid syndrome [44].

The diagnosis of carcinoid heart disease is established by a transthoracic echocardiography carried out by an expert in carcinoid heart disease for SI-NETs. As most patients are initially asymptomatic, the European Neuroendocrine Tumor Society (ENETS) recommends a transthoracic echocardiography in patients with elevated urinary 5-HIAA levels regardless of the presence of carcinoid syndrome [14]. A urinary 5-HIAA level ≥ 300 μmol/24 h is an independent predictor of the development or progression of carcinoid heart disease [46].

It has been shown that cardiac neurohormone N-terminal pro-brain natriuretic peptide (NT-pro-BNP) has both diagnostic significance with a sensitivity of 92% and a specificity of 91% for a cut-off level of 260 pg/mL (31 pmol/L) and prognostic significance for carcinoid heart disease [47,48]. A major elevation of NT-pro-BNP levels should strengthen the indication of preoperative transthoracic echocardiography.

If carcinoid heart disease is diagnosed, cardiological management must be carried out before any oncological surgery [44]. Cardiological management requires pharmacotherapy of heart failure (e.g., diuretics) and somatostatin analogues in order to control carcinoid syndrome and prevent the progression of carcinoid heart disease. In cases of severe valve regurgitation, emergent valve surgery is required before any abdominal surgery.

### 3.3. Prevention and Management of the Carcinoid Crisis

Carcinoid crisis seems to be the continuation of carcinoid syndrome even if its physiopathology remains largely unknown [49]. Carcinoid crisis is an acute, life-threatening condition which can occur during any kind of invasive procedure in SI-NET patients. This crisis can combine severe hemodynamic instability, cardiac arrhythmia, cardiac failure, and refractory bronchoconstriction. If prolonged, it may also increase the postoperative morbidity rate [50,51]. The two main risk factors identified for carcinoid crisis are the presence of liver metastases and a history of carcinoid syndrome [50,52]. Yet, such crises have been observed during procedures in SI-NET patients without identified risk factors and physicians should always be prepared to manage them [52].

Even if octreotide is the treatment of choice for carcinoid crisis according to its efficacy in carcinoid syndrome, data on its value for both managing and preventing the crisis are still scarce and conflicting [50,51,52,53,54,55,56]. All of the existing data are based on retrospective studies which used heterogenous definitions of carcinoid crisis, primary tumor origins, stages of disease, and octreotide regimens. In 2001, Kinney et al. published the first and only study with a control group: they reported no complications during surgery in patients treated with octreotide (0/45, 0%) preoperatively, unlike those who did not receive any octreotide (7/67, 10%) [53]. The other publications, without any control group, described a low rate of carcinoid crisis (range from 0% to 3.4%) with the systematic use of octreotide regimen on the one hand, and a high rate of carcinoid crisis (range from 24% to 32%) despite the systematic use of octreotide regimen [50,51,52,54,55,56] on the other hand. Whatever the efficacy of octreotide regarding carcinoid crisis, it does not seem deleterious, as no increased rate of anastomotic leakage has been reported despite the octreotide-induced decrease of visceral perfusion. Consequently, octreotide prophylaxis must be included within the perioperative management protocol at a rate ranging from 100–500 mcg/h, even more so if risk factors are present [15,25]. Despite this preparation, anesthesiologists must be ready to face a carcinoid crisis by prompt usage of octreotide bolus but also vasopressor drugs (such as vasopressin and phenylephrine) in order to limit the duration of hypotension, which could lead to postoperative morbidity.

## 4. Surgery

Surgical treatment is a cornerstone of SI-NET multimodal management and may be curative when a complete R0 resection is obtained. However, a R0 resection is feasible in only 20% of cases due to the usual advanced stage at diagnosis [57].

### 4.1. Surgery of Primary Tumor and Mesenteric Lymph Node Metastases in Curative Intent

Surgery is the gold standard for curative treatment of SI-NETs. The main rules for SI-NET surgery are the resection of all the primary tumors, associated with a systematic lymphadenectomy, while focusing on the preservation of the small bowel [14,15]. Short bowel syndrome should be avoided. The length of the resected small bowel is not correlated to the amount of resected lymph nodes [29,57]. The surgical approach should not be a ‘pizza pie’ approach—large intestinal plus small mesentery resection—and the inverse approach should become the standard (Figure 3) [25]. It may be advisable for such interventions to be performed at specialized centers.

#### 4.1.1. Exploration of the Abdomen

Exploration is the first surgical step, as up to 60–70% of the disease is missed on preoperative imaging [17,57,58]. The surgeon must palpate the entire small bowel and search for carcinomatosis and ovarian and liver metastases.

Manual palpation of the entire small bowel maximizes the detection of multiple synchronous tumors, missed in more than 60% of patients on preoperative imaging [6,59,60]. However, palpation still remains imperfect, as more lesions are discovered by pathology, assuming that small primary tumors may be left in place [17]. After palpation, the surgeon puts a mark on the most proximal and distal palpated tumor to define the extent of bowel resection for oncological purposes.

The liver surface should also be examined, as half of the number of liver metastases are undetected on preoperative imaging [34]. Ultrasound can be a good complement to palpation and inspection of the liver surface. However, more metastases are still found on pathology than during peroperative examination and this is even truer if considering micro-metastases [32,33,34].

#### 4.1.2. Surgery of MLNM

Lymphadenectomy is the second surgical step and should be performed before the bowel resection. The remnant vascularized intestine after mesentery dissection will guide the extent of bowel resection for vascular purposes (Figure 4). Lymphadenectomy may be challenging, especially in the presence of extensive mesenteric fibrosis or large MLNM surrounding the superior mesenteric vasculature (LN stage III). Lymphadenectomy is always required, even in cases of small primary tumors (<1 cm), because MLNM are almost always present (80%) (Table 1), and because lymphadenectomy improves survival and may avert acute local complications [3,7,9,17,25,61]. Despite consensual agreement about the need for lymphadenectomy, up to 20% of patients do not have any lymph node resection during SI-NET surgery in the largest published series [7,8]. Surgeons must be informed that lymphadenectomy is always required in SI-NET surgery.

Retrospective registry analyses have suggested that at least 8 (or possibly 12) removed lymph nodes are needed to improve overall survival [7,8,9]. Moreover, French guidelines have suggested discussing a ‘re-intervention’ after postoperative evaluation by FDOPA-PET or ^68^Ga-PET if fewer than 8 lymph nodes have been resected [25]. This is most frequently observed after emergency surgery.

The upper limit of lymphadenectomy is less consensual. In the absence of a retro-pancreatic target on preoperative imaging, lymphadenectomy is usually conducted along the trunk of the superior mesenteric vessels below the pancreas. However, Pasquer et al. reported the presence of lymph node skip metastases (in 14/21, mainly metastatic patients) and suggested that a systematic lymph node dissection up to the retro-pancreatic area should be realized [59]. Nevertheless, the benefit/risk ratio of such a procedure in patients without liver metastases should be demonstrated, because of the potential morbidity of extensive lymphadenectomy [59]. Lymphatic mapping may be an interesting way to fix the limits of the lymph node harvest using isosulfan/methylene blue, infrared fluorescent lymph node navigation or radio-guided but this practice is not yet performed as standard nor it is recommended [62,63,64]. Bowel resection is generally not a key issue. The exact procedure depends on (i) the number of palpated primary tumors, (ii) their exact location generally in the last portion of the ileum, and (iii) the remnant vascularized intestine after mesentery dissection. For proximal SI-NETs, a small bowel enterectomy can be a good option in order to keep in place the ileocecal valve and limit intestinal symptoms. However, a right hemicolectomy is required in most patients because tumors are located in the terminal ileum and/or the right colic artery is resected. Whatever the procedure, the maximum length of small bowel must be preserved to preclude malabsorption and diarrhea, especially bile-salt-induced diarrhea [65]. Despite every precaution, if patients have postoperative diarrhea, they may require either medical (e.g., cholestyramine) or nutritional support [14].

#### 4.1.3. Emergency Surgery

Most patients with SI-NETs (nearly 80%) present nonspecific symptoms mainly related to the local effect of MLNM than to the primary tumor [4,5,66]. Among them, 12.5% to 33% need emergency surgery [5,12,67]. Acute symptoms requiring emergency surgery are small bowel obstruction (80%), pain (10%), and less frequently mesenteric ischemia, intussusception or gastrointestinal bleeding [5,12]. Emergency surgery raises some questions, including (i) a higher risk of postoperative complications, (ii) an inadequate surgical act (lack of lymphadenectomy, multiple primary tumors left in place, or extensive small bowel resection), and (iii) an earlier relapse [5,12]. In order to limit those risks, patients should be referred to specialized centers [68], but an oncologic resection may not be possible in an emergency context. In this case, the procedure should focus on the life-threatening condition, and consider a limited resection of the diseased intestine to avoid subsequent difficult reoperation [25]. Thereafter, a re-intervention for oncological purposes should be planned if the first emergency surgery was non-optimal R2 resection, fewer than eight resected lymph nodes, absent of entire small bowel palpation, carcinomatosis, and positive postoperative imaging [25].

### 4.2. Surgery of Metastases in Curative Intent

#### 4.2.1. Curative Surgery for Liver Metastases

Nearly 50% of patients with SI-NETs present liver metastases at diagnosis [10]. When feasible, their radical destruction is the only potential curative treatment and is the standard of care even though this strategy has never been compared properly with other treatments [35,69]. Radical destruction can be performed by resection (metastasectomy, partial hepatectomy, liver transplantation) alone and/or associated with percutaneous or intraoperative thermal ablation (radiofrequency and/or microwave). The destruction of the liver metastases can be safely combined to the resection of the primary tumor plus mesenteric lymphadenectomy [70].

Each candidate for radical destruction should be discussed at a multidisciplinary meeting. ENETS guidelines recommend considering radical destruction if it can be achieved with an acceptable predicted morbidity <30% and mortality rate <5% [71]. Selection criteria are patients with (i) type I or at least type II liver metastases, (ii) stable disease, (iii) no extra abdominal metastatic disease on dedicated metabolic imaging, (iv) a good performance status, and (v) no carcinoid heart disease [25,68,69]. In patients with type II liver metastases, a two-step approach, which includes resection of left metastases associated with a right portal vein ligation followed by right hepatectomy, may be proposed [69,72].

Radical treatment of liver metastases could be more palliative than curative: even if radical liver treatment is achieved, recurrence is the rule when the follow-up is long enough (Table 2) [34,73,74]. The rate of relapse may be explained by the existence of micro-metastases in nearly all patients with macro-metastases [32,33,34].

Regarding their relatively low aggressive behavior, SI-NET liver metastases can be an accepted indication for liver transplantation for selected patients with unresectable liver disease, without any other organ involvement and with pre-transplant curative resection of all extrahepatic lesions; but the patient selection criteria remain vague superior mesenteric artery [81].

Recent publications reported a five-year overall and disease-free survival after liver transplantation respectively between 47% and 71% and 31% and 57% [81]. Those data suggest that liver transplantation is also more of a palliative than a curative option.

#### 4.2.2. Peritoneal Carcinomatosis

Peritoneal carcinomatosis affects approximately 20% of patients with SI-NET [11]. Peritoneal carcinomatosis is described as an independent poor-prognosis factor, and although it is not immediately fatal, it worsens quality of life in about one-fifth of patients [3,82]. When feasible, its complete surgical resection is admitted as the only potential curative treatment. Peritoneal carcinomatosis surgery may also improve the patient’s prognosis and avoids local complications such as chronic occlusion and pain [15,68,83,84,85]. PRRT may be irrelevant in this indication: Merola et al. noted a lack of disease control in nearly 40% of patients, and the occurrence of complications such as bowel obstruction and/or ascites (maybe due to radiation-induced peritonitis or paralytic ileus) in about 30% of the treated subjects [82].

Perioperative scores are commonly used to evaluate the extent and the resectability of peritoneal carcinomatosis. Among the available scores, the Peritoneal Carcinomatosis Index (PCI) is the most frequently used even if it has not yet been validated in SI-NETs. A PCI score >20 may lead to a failure of radical resection [86,87]. To consider the disease as a whole, ENETS proposed the gravity peritoneal carcinomatosis score (GPS) which takes into account not only peritoneal carcinomatosis resectability but also the other location of the disease; this score has not yet been evaluated prospectively [83]. They suggested avoiding peritoneal surgery in GPS-C patients (patients with peritoneal carcinomatosis and major liver involvement and/or extra abdominal lymph node metastases).

If the indication for complete resection of the peritoneal disease is admitted, the added value of hyperthermic intraperitoneal chemotherapy (HIPEC) remains unknown. The largest published experience of HIPEC in SI-NETs was published by Elias et al. [85]. Elias et al. stopped using HIPEC in complement of surgical resection after the treatment of 28 SI-NET patients because of the associated induced morbidity and the absence of difference in overall survival in the HIPEC group [85]. To date, NANETS and French guidelines do not recommend the use of HIPEC in SI-NETs [15,25]. In highly-selected fit patients with predominant peritoneal disease, HIPEC may complete the cytoreduction, but there is a need for further studies.

#### 4.2.3. Palliative Surgery

Palliative strategy in SI-NET patients aims to relieve symptoms and delay a fatal outcome which is mainly due to liver failure from liver metastases and otherwise bowel complications. Cytoreductive surgery is part of the multimodal palliative strategy even if its position and its benefits have not been unequivocally proven by randomized control trials.

##### Resection of the Local Disease (Primary Tumor + MLNM) when Liver Metastases are Unresectable

Resection of symptomatic local disease (nearly 80% of cases) is recommended in order to relieve symptoms in the setting of unresectable liver metastases [15,68,88]. However, the resection of asymptomatic local disease is debated in the context of unresectable liver metastases. To date, only retrospective data are available concerning this issue [89,90].

Arguments supporting resection in this context are: (i) the prevention of local complications, (ii) the control of locoregional disease to focus on hepatic treatment, and (iii) the improvement of overall survival rate. A recent meta-analysis favored asymptomatic local disease resection with a better five-year survival rate in the resected vs. the non-resected group (36.6% and 73.1%, respectively), and this resection seemed to be safe (<2%, 30-day mortality) [88]. In spite of these arguments, Daskaslis et al. found no survival advantage for local disease resection in asymptomatic stage IV SI-NET patients after propensity matching (91 patients in each group) [90]. In this study, however, more than half of the patients in the non-resected group underwent surgery, and this might have impacted survival results [90]. Results from meta-analyses supporting palliative surgery must be weighed against available high-level evidence from randomized trials (PROMID, CLARINET, RADIANT-4 or NETTER-1) that showed long-term survival in metastatic patients receiving systemic therapies [15].

Most academic guidelines (ENETS, UK and Ireland Neuroendocrine Tumour Society (UKINETS), North American Neuroendocrine Tumor Society (NANETS), *Thésaurus National de Cancérologie Digestive* (TNCD)) recommend local disease resection in cases of asymptomatic localized tumor and unresectable liver metastases, but not the National Comprehensive Cancer Network guidelines [25,69,91,92]. To date, either the resection or the non-resection of asymptomatic primary tumors in the context of unresectable liver metastases are acceptable, but each case should be discussed in a multidisciplinary meeting. Life-threatening MLNM, albeit a subjective definition, should be resected before they become too bulky.

##### Palliative Surgery for Inextirpable Bulky MLNM

MLNM plus fibrosis can be unresectable when they surround the origin of the mesenteric vessels (LN stage IV). Symptoms related to these bulky MLNM vary among patients, including absence of symptoms, chronic mesenteric ischemia, and bowel obstruction. Patients, who are asymptomatic, with the development of a collateral circulation avoiding mesenteric ischemia, are treated medically. If they are symptomatic, the optimal strategy in order to relieve symptoms is aggressive surgery—radical or partial debulking surgery (Figure 5) preserving the first jejunal arteries and thus the small bowel vascularization [15,68,88,90,93,94]. Such patients should be evaluated and operated on in specialized centers [68].

In a few patients, debulking surgery cannot be performed or it does not relieve symptoms, especially ischemic ones. In such cases, the insertion of a self-expandable stent in the superior mesenteric vein (more easily compressed than the artery) at the level of the mass, inserted through the portal vein, has been undertaken in a small number of patients with uncertain results [90,95].

##### Palliative Debulking Surgery for Liver Metastases

In patients with unresectable liver metastases, palliative surgery and/or thermal ablation are just part of the multimodal strategy including arterial embolization, chemoembolization, PRRT, and liver transplantation. Despite the absence of evidence through randomized control trials, palliative liver surgery may be proposed to decrease local or hormonal symptoms, and possibly to improve survival rates. It has been supported that 90%, or more recently >70%, of the metastatic liver burden should be resected in order to obtain clinical benefit [90,93,94,95,96].

### 4.3. Open vs. Laparoscopic Resections

Laparotomy is traditionally the approach of choice, because it allows an optimal exploration of the entire abdominal cavity including small bowel palpation and vascular control at the origin of the superior mesenteric vessels [68]. For these reasons, a purely laparoscopic approach for curative surgery in SI-NETs remains controversial [15,57,58]. Hence, NANETS and ENETS guidelines suggest a hybrid procedure combining laparoscopy and manual palpation of the entire small bowel after its exteriorization by a hand port (Figure 6), as proposed by Wang et al. [15,68,97]. Yet, such a procedure should not be considered if an incomplete nodal resection is predictable large MLNM or MLNM around the superior mesenteric axis [57]. We also recommend the exclusion of patients with retropancreatic MLNM, carcinomatosis, and/or abdominal obesity which may preclude intestinal exteriorization by the hand port.

In palliative intent, the laparoscopic approach seems to be advantageous for the resection of the local disease in patients with unresectable liver metastases.

### 4.4. Prophylactic Cholecystectomy

In SI-NET patients, the prevalence of gallstone disease ranges between 36% and 63%, and the five-year cumulative risk of having cholecystectomy or a biliary drainage is nearly 20%, which is considerably higher than in the general population [98,99,100,101]. The identified risk factors for cholelithiasis in SI-NET patients are firstly the treatment by somatostatin analogues, and secondly previous ileal resection [98,102]. Furthermore, SI-NET patients are also exposed to the risk of ischemic cholecystitis after trans-arterial embolization for liver metastases therapy [103].

Cholecystectomy at the time of the SI-NET surgery is recommended by NANETS and ENETS guidelines if long-term somatostatin analogue treatment is anticipated [14,15]. This additional procedure does not seem to increase the morbidity and mortality rate of the intervention [104]. More generally, concomitant cholecystectomy should always be discussed and the final decision depends on technical aspects (emergency vs. elective surgery, expected intraoperative risk of cholecystectomy) and clinical aspects (presence of cholelithiasis, history of symptomatic cholelithiasis, planned somatostatin analogue treatment or trans-arterial embolization) [14,101].

### 4.5. Place of Neoadjuvant Treatment

Downsizing SI-NET lesions is an attractive concept, to make a non-resectable tumor resectable or to facilitate the resection of a borderline lesion (LN stage III or IV and type II or III liver metastases). Unfortunately, neoadjuvant treatments are lacking in SI-NETs as they are poorly chemo-sensitive (either with cytotoxic or targeted chemotherapy) and as somatostatin analogues may increase survival but do not shrink the tumor. Regarding PRRT, the NETTER-1 trial reported a partial response according to RECIST in 18% of SI-NET patients with well differentiated G1/G2 advanced tumors, progressing on long-acting octreotide [105]. So far, only a few case reports provide conflicting data on the neoadjuvant use of PRRT, and further studies are needed [106].

## 5. Postoperative Management

### 5.1. Postoperative Evaluation

Several studies have reported a long-term recurrence rate of around 50% after curative R0 surgery for SI-NET without distant metastases [5]. There is no consensus on the optimal duration of the postoperative follow-up, but on the basis of their slow growth, metastatic recurrences may occur several years after surgery, and surveillance should be continued at least 20 years hence or even lifelong, especially in young patients or in those with high risk of recurrence [15].

After surgery in curative intent, the ENETS consensus guidelines and French guidelines recommend performing conventional imaging (CT or MRI with diffusion-weighted imaging), functional imaging (using the previous positive scintigraphic technique), and chromogranin A and 5-HIAA level measurements at regular intervals: from 3–6 months initially, then every 6–12 months over the following five years, then every 12–24 months over the following ten years, and finally every five years [25,107]. Exceptional G3 tumors should be monitored every three months. MRI with diffusion-weighted imaging is non-ionizing and is preferred for the detection and/or the follow-up of liver metastases [30]. Hence, abdominal MRI (for the liver) and CT scans (for extra-hepatic lesions) may be used alternately. Caution is recommended when comparing the results of different imaging techniques (higher sensitivity of the ^68^Ga-DOTA PET/CT compared with OctreoScan, and MRI compared with CT scan for liver lesions). As described above, chromogranin A reflects tumor burden and may be used to assess the speed of tumor growth. Some series have shown that elevated chromogranin A levels can be observed months to years before the evidence of radiographic recurrence, suggesting its usefulness for the early detection of tumor relapse after successful curative resection [107]. However, these data are contradictory [25]. Twenty-four hour urinary 5-HIAA level measurements may be used to detect and evaluate carcinoid syndrome or carcinoid heart disease. Plasma 5-HIAA levels may be easier to measure but their use is not currently recommended. NT-pro-BNP levels can be monitored to promote early detection of cardiac involvement during follow-up.

In the case of unresected liver metastases, CT or MRI and biochemical follow-up should be performed at three months, then every 3–6 months for two years, and finally every 6–12 months in cases of stable disease [25,107].

### 5.2. Adjuvant Treatment

After curative surgery for SI-NETs, there is no proven role for adjuvant systemic therapy [14].

Somatostatin analogues are indicated for their antiproliferative efficacy as first-line therapy in cases of slowly progressive well-differentiated metastatic SI-NETs, or for antisecretory purposes in cases of persistent secreting syndrome or refractory diarrhea through inhibition of gastrointestinal secretions [41,42]. They have not been studied as an adjuvant therapy to prevent SI-NET relapse after curative surgery.

Regarding PRRT, in the NETTER-1 trial, PRRT improved progression-free survival by 79% (*p* < 0.00001) compared to high-dose octreotide in patients with SI-NETs progressing radiographically on somatostatin analogue therapy) [105]. To date, PRRT is not recommended as an adjuvant therapy after curative surgery for SI-NET, but a randomized phase III French study (TERAVECT) is being conducted to evaluate the interest of PRRT with In111-pentetreotide in consolidation after complete resection of liver metastases in patients with well-differentiated digestive neuroendocrine tumors.

No adjuvant chemotherapy is indicated after curative surgery for SI-NETs.

## 6. Conclusions

SI-NET surgery presents specific challenges but offers the opportunity to provide significant improvements in both quality of life and survival in SI-NET patients. Figure 7 proposes an algorithm diagram for surgical indications of resection for SI-NET local disease. As SI-NETs have an increasing frequency, surgeons must be informed about the specific surgical management according to their hormonal secretion, their usual dissemination at the time of the diagnosis and the need for bowel-preserving surgery to avoid short bowel syndrome.

## Figures and Tables

**Figure 1 jcm-09-02319-f001:**
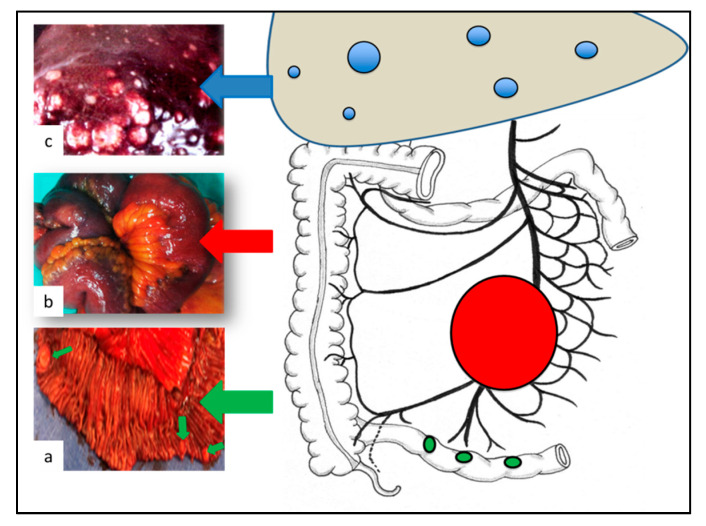
Classical presentation of small-intestinal neuroendocrine tumors (SI-NETs). (**a**) Small primary tumors (<20 mm), distal in the ileum, and multiple in 30% to 50% of cases; (**b**) Mesenteric lymph node metastases present in more than 80% of cases at diagnosis, typically larger than primary tumors; (**c**) Liver metastases present in approximately 50% of cases at diagnosis usually multiple and bilobar.

**Figure 2 jcm-09-02319-f002:**
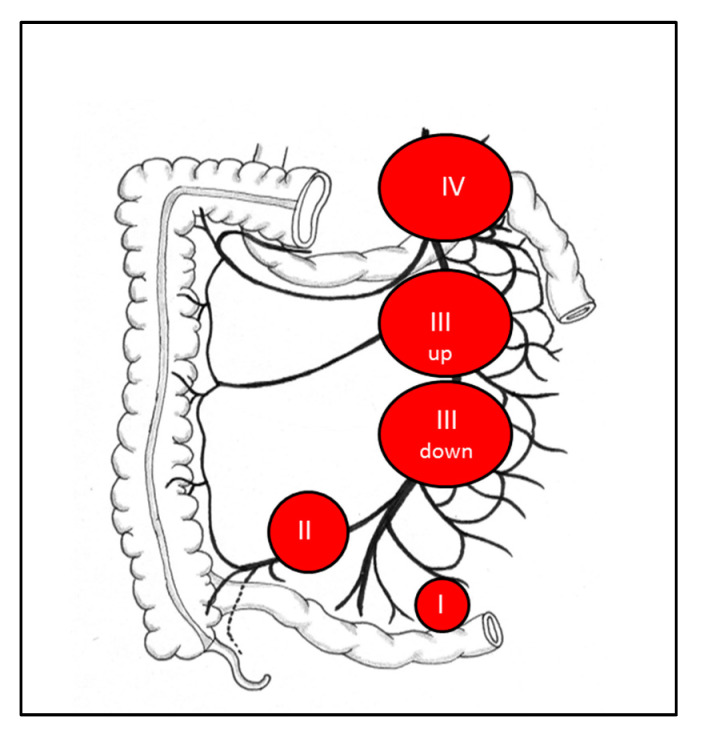
Classification based on the relationship between mesenteric lymph node metastases (MLNM) and the superior mesenteric vessels. Lymph node stage 0: no visible mesenteric MLNM suspicious of malignancy; Lymph node stage I: proximity to the small intestine, without invasion of the superior mesenteric artery lymph node (LN); Lymph node stage II: involvement of the distal branches of the superior mesenteric artery (SMA), next to their origin; Lymph node stage III: involvement of the trunk of the SMA, without involving the first jejunal arteries; III ‘up’: <3–4 free jejunal branches; III ‘down’: >3–4 free jejunal branches; Lymph node stage IV: involvement of the trunk of the SMA with involvement of the first jejunal arteries.

**Figure 3 jcm-09-02319-f003:**
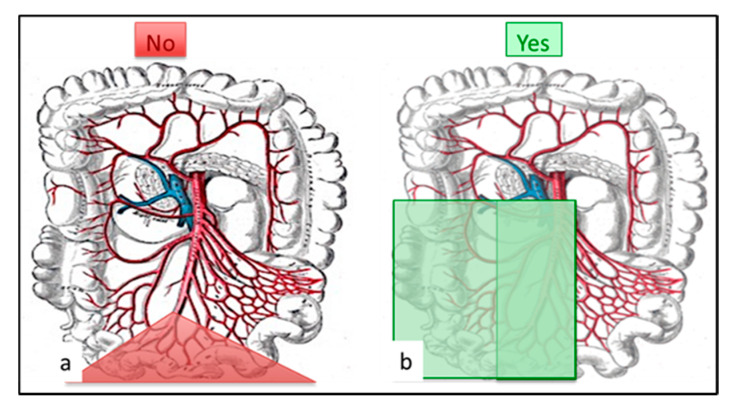
Surgical pattern: (**a**) no more ‘pizza pie’ approach: large intestinal resection with inappropriate lymphadenectomy. (**b**) Appropriate lymphadenectomy should remove at least 8 (better 12) lymph nodes with small intestinal resection. The ileocecal valve and right colon might require resection mainly for LN-stage III patients.

**Figure 4 jcm-09-02319-f004:**
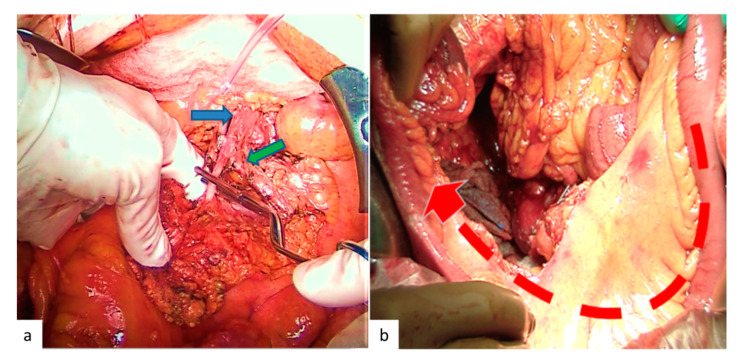
Operative view during the lymphadenectomy. (**a**) Clamping test after the dissection of the mesenteric superior artery and before the resection of the small bowel to visualize the remnant vascularized bowel. The blue arrow shows the mesenteric superior artery, and the green arrow shows the firsts jejunal arteries. (**b**) Small bowel vascularized by the remnant jejunal arteries.

**Figure 5 jcm-09-02319-f005:**
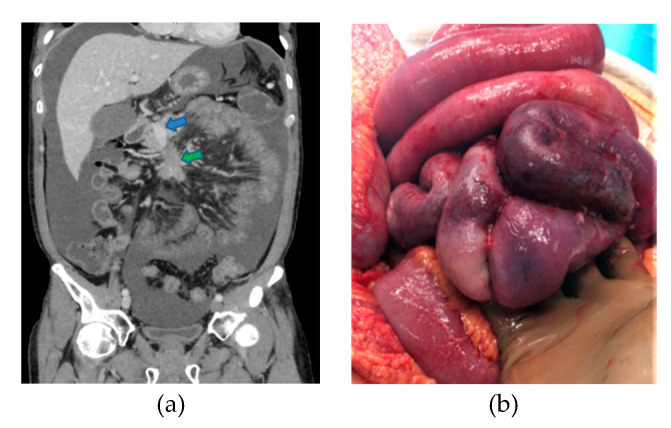
Mesenteric ischemia due to bulky MLNM with encasement of the mesenteric vessels. (**a**) Computed tomography (CT) scan: mesenteric superior venous stenosis (blue arrow) and MLNM (green arrow). (**b**) Debulking surgery for mesenteric ischemia due to mesenteric vessels encasement by MLNM.

**Figure 6 jcm-09-02319-f006:**
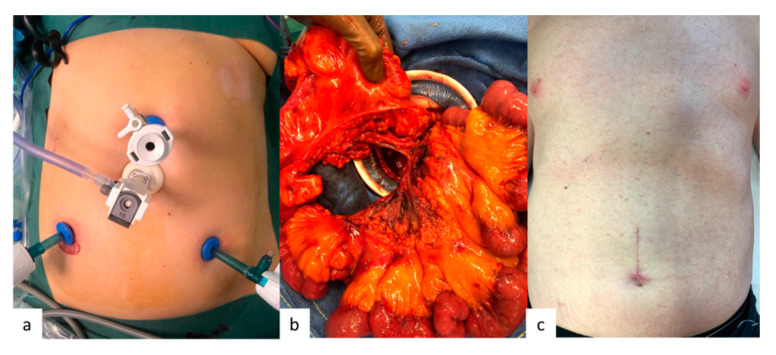
Operative view of a hybrid procedure combining laparoscopy and manual palpation of the entire small bowel. (**a**) Laparoscopic preparation; (**b**) Exteriorization of all the entire small bowel, the right colon, and the mesentery; (**c**) Postoperative scarce.

**Figure 7 jcm-09-02319-f007:**
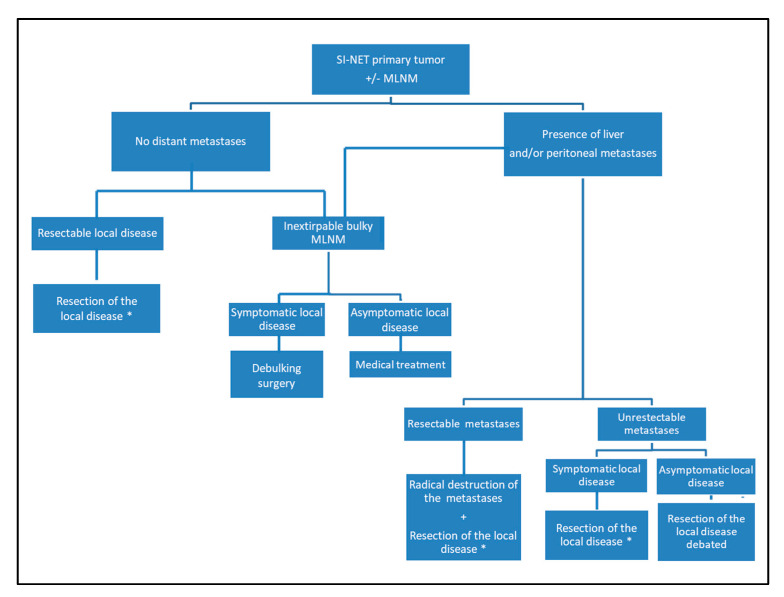
Proposed algorithm diagram for surgical indications of resection for SI-NET local disease. * Resection of all the primary tumors (after manual palpation of the entire small bowel) + systematic mesenteric lymphadenectomy (with at least 8 or 12 removed LN).

**Table 1 jcm-09-02319-t001:** Rate of MLNM in SI-NET.

Author	Study Period	Patients (n)	Disease Stage	Presence of MLNM * (%)	Presence of MLNM When Primary Tumor <1 cm * (%)	Patients Without Any Lymph Nodes Resection (%)
Chen [9]	2004–2014	1925	Stage I-III	80.3	-	-
Landry [7]	1997–2004	1364	Stage I-IV	82	-	16.2
Motz [8]	1998–2013	11,852	Stage I-III	79.3	46.7	19.2
Norlén [3]	1985–2010	517	Stage I-IV	93	-	-

* when at least one lymph node was removed.

**Table 2 jcm-09-02319-t002:** Recurrence after radical liver surgery for neuroendocrine tumor (NET) liver metastases.

Author	Year	Patients (*n*)	Length of Follow-Up (Years)	Relapse (%)
Chen [75]	1998	3	5	67
Chamberlain [76]	2000	28	5	89
Jaeck [77]	2001	4	3	31
Sarmiento [73]	2003	90	10	94
Elias [78]	2003	14	10	89
Kianmanesh [72]	2008	23	4	48
Scigliano [79]	2009	41	5	78
Bertani [80]	2015	78	8	81

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
