# Peer review of "Surgery and Perioperative Management in Small Intestinal Neuroendocrine Tumors"

_jcm, 2020, doi:10.3390/jcm9072319_

Round 1

Reviewer 1 Report

In their manuscript entitled "Surgery and perioperative management in small intestinal neuroendocrine tumors" the authors present a narrative yet thorough review of surgical treatment options available for SI-NETs.

The title accurately reflects the manuscript content, the text is well-structured and references are up-to-date. The provided figures are illustrative and explanatory to the readership.

Overall this is a very carefully presented work and is of somewhat significant clinical value for clinicians.

Some minor points that could improve the present work include:

  • The authors could also add available data on synchronous metastatic and primary SI-NET resections or resection +ablation.
  • When referring to OLT for unresectable NELMs the authors could provide with some published survival outcomes.
  • An algorithm diagram for clinical decision making could be added.

Author Response

Dear reviewer

Thank you for your remarks. We tried hard to bring all the request changes.

  • The authors could also add available data on synchronous metastatic and primary SI-NET resections or resection +ablation.

à The destruction of the liver metastases can be safely combined to the resection of the primary tumor plus mesenteric lymphadenectomy (addeo).

  • When referring to OLT for unresectable NELMs the authors could provide with some published survival outcomes.

à Regarding to their relatively low aggressive behaviour, SI-NET liver metastases can be an accepted indication for liver transplantation for selected patients with unresectable liver disease, without any other organ involvement and with pre-transplant curative resection of all extrahepatic lesion. But the patient selection criteria remain vague.

Recent publications reported a 5-year overall and disease-free survival between 47 to 71% and between 31 to 57%, respectively from liver transplantation for NET liver metastases (kim). Those data suggest that liver transplantation is also more a palliative than a curative option.

  • An algorithm diagram for clinical decision making could be added à it has been added (Figure 7)
    • The authors could also add available data on synchronous metastatic and primary SI-NET resections or resection +ablation.

    à The destruction of the liver metastases can be safely combined to the resection of the primary tumor plus mesenteric lymphadenectomy (addeo).

    • When referring to OLT for unresectable NELMs the authors could provide with some published survival outcomes.

    à Regarding to their relatively low aggressive behaviour, SI-NET liver metastases can be an accepted indication for liver transplantation for selected patients with unresectable liver disease, without any other organ involvement and with pre-transplant curative resection of all extrahepatic lesion. But the patient selection criteria remain vague.

    Recent publications reported a 5-year overall and disease-free survival between 47 to 71% and between 31 to 57%, respectively from liver transplantation for NET liver metastases (kim). Those data suggest that liver transplantation is also more a palliative than a curative option.

    • An algorithm diagram for clinical decision making could be added à it has been added (Figure 7)

Best regards

Sophie Deguelte

Reviewer 2 Report

Deguelte et al review the surgical and perioperative management of the small-intestinal neuroendocrine tumors (SI-NET). Which are the most prevalent small bowel neoplasms with an increasing frequency. Surgery plays a key role in the multidisciplinary management of SI-NET; both when the intent is curative intent, and if the fails also later when the aim is palliative. The paper gives a very well-balanced overview of the surgical indications and techniques, and perioperative and postoperative management of SI-NETs balancing the different societies slightly different guidelines.

The review is very welcome in the constantly evolving field and I have not further comments.

Author Response

Dear Reviewer

Thank you for your remarks.

Best regards

Sophie Deguelte

Reviewer 3 Report

This is very well structured and clearly written review. I have minor comments:

  1. Word “cases” repeats too many times throughout the whole manuscript, it is better not to use “cases” when you speak about patients.
  2. Figure 4, picture a) contains two arrows. When you read the description below it is not clear to what these arrows are pointing – superior mesenteric artery, first jejunal branches? Please clarify this.
  3. Line 329 – please split the word furtherstudies
  4. Line 330 – Palliative urgery, letter "s" is missing
  5. Near infrared fluorescent lymph node navigation is being extensively used for lymph node mapping, could you please comment on this in your paper.

Author Response

Dear Reviewer,

Thank you for your remarks. We made the requested modifications.

  1. Word “cases” repeats too many times throughout the whole manuscript, it is better not to use “cases” when you speak about patients. --> done
  2. Figure 4, picture a) contains two arrows. When you read the description below it is not clear to what these arrows are pointing – superior mesenteric artery, first jejunal branches? Please clarify this.

The blue arrow shows the mesenteric superior artery, and the green arrow shows the firsts jejunal arteries.

  1. Line 329 – please split the word furtherstudies --> done
  2. Line 330 – Palliative urgery, letter "s" is missing -->done
  3. Near infrared fluorescent lymph node navigation is being extensively used for lymph node mapping, could you please comment on this in your paper. --> done

best regards

Sophie Deguelte